# Validation of the Brief Thai Version of the Work-Related Quality of Life Scale (Brief THWRQLS)

**DOI:** 10.3390/ijerph17051503

**Published:** 2020-02-26

**Authors:** Teerayut Kongsin, Naesinee Chaiear, Nikom Thanomsieng, Sirintip Boonjaraspinyo

**Affiliations:** 1Division of Occupational Medicine, Department of Community Medicine, Faculty of Medicine, Khon Kaen University, Khon Kaen 40002, Thailandsboon@kku.ac.th (S.B.); 2Occupational Health and Safety Office, Faculty of Medicine, Khon Kaen University, Khon Kaen 40002, Thailand; 3Department of Biostatistics and Demography, Faculty of Public Health, Khon Kaen University, Khon Kaen 40002, Thailand

**Keywords:** quality of work-life, short form, validity, reliability, health personnel

## Abstract

The Work-Related Quality of Life Scale (WRQLS) was developed by Van Laar et al. The Thai version was developed and could be completed in 13.4 min on average with some items having a factor loading of less than 0.4. The aims of this study were (a) to develop a brief Thai version of the WRQLS (brief THWRQLS), and (b) to assess its validity and reliability. A descriptive correlation study was performed with the components of THWRQLS selected based on statistical and judgmental criteria. The statistical criteria were developed using secondary data from 320 physicians identifying discrimination, internal consistency and exploratory factor analysis (EFA). The judgmental criteria included content validity and agreement by five experts. The web-based brief THWRQLS was then used by 250 health personnel, and confirmatory factor analysis (CFA) was conducted and internal consistency assessed. The brief THWRQLS consisted of seven dimensions, encompassing 25 of the original 32 items. The CFA revealed that most of the standardized factor loadings were greater than 0.5. The χ^2^goodness of fit was 268.772 (*p* < 0.01), the comparative fit index was 0.971, the root mean square error of approximation was 0.039, and the standardized root mean square residual was 0.049. The Cronbach’s alpha coefficient of the scale was 0.94, and almost all dimensions were greater than 0.7 except for that of “stress at work,” which was 0.53. In conclusion, the brief THWRQLS appeared to be valid, and the reliability was acceptable, except in the dimension of “stress at work.”

## 1. Introduction

Quality of work-life refers to all of the organizational inputs that aim to enhance employee satisfaction and organizational effectiveness [1]. According to Walton, the idea of the quality of work-life, which developed during the 1970s, includes the values “human needs and aspirations”, “adequate and fair compensation for work”, and “social relevance of work” [2].

Most organizations encounter psychosocial stressors, especially hospitals. Healthcare professionals suffer from various psychosocial stressors in their workplaces, such as role conflict, emotional labor, being concerned about medical errors and litigation, as well as experiencing verbal or physical abuse by patients and caregivers or bullying by colleagues [3,4]. In addition, healthcare professionals have a much greater chance of being exposed to long working hours, night work, or shift work and conflict among co-workers. These stressors affect the quality of work-life and can lead to burnout, depression, anxiety disorders, sleep disorders, or other psychiatric disorders [5,6,7,8,9].

Tools for assessing the quality of work-life should have good psychometric properties. Initially, before the development of a questionnaire for evaluating the quality of work life, quality of life (QoL) was used as a proxy to assess the quality of work-life [10,11,12]. The 23-item Work-related Quality of Life Scale (WRQLS), which was first developed in England by Van Laar et al., was developed based on six factors: job and career satisfaction (JCS), general well-being (GWB), home–work interface (HWI), stress at work (SAW), control at work (CAW), and working conditions (WCS). Its validity and reliability have been established among medical personnel in the United Kingdom [13], and it has been used in various types of organizations and translated into 13 languages (including Thai) [14]. The WRQLS-2 was a modified version of the WRQLS that initially consisted of six factors and 34 items, that was later adjusted to comprise seven factors, adding employee engagement EET) and 32 items. According to Sirisawasd et al., the reliability (Cronbach’s α of 0.671–0.82) and validity (Cronbach’s α of 0.92) of the Thai version of the tool (THWRQLS) was sufficient to assess the quality of work-life among nurses in Thailand. However, the THWRQLS is time-consuming, taking an average of 13.4 min to complete (SD = 4.58 min, range 8–20 min) [15]. In addition, a seven-factor model generated from the principal component factor analysis found that five items have factor loadings of less than 0.4, and Somsila et al. [16] found that the content of some items overlapped. This suggests that the original 32-item scale contains several items that tend not to perform well psychometrically.

Questionnaires provide for a large amount of people or a time-limited organization, but the content should be concise, easily understood and contain a suitable number of items that still retain good psychometric properties [17]. Therefore, the objectives of this research were to develop a brief Thai version of the work-related quality of life scale (brief THWRQLS) that would reduce the amount of content without sacrificing measurement precision, and to assess its validity and reliability.

## 2. Materials and Methods

### 2.1. The Selection of Components to be Included in the Brief THWRQLS

The brief Thai version of the work-related quality of life scale (Brief THWRQLS) was developed based on the items included in the original 32-item scale. The selection of items to be included was based on expert consensus using data based on statistical and judgmental criteria. The statistical criteria were developed based on statistical analysis of secondary data obtained by Soonthornvinit et al. [18] in a study on the quality of work-life among doctors in the northeastern region of Thailand. In the factor analysis literature, the minimum sample size necessary to obtain factor solutions that are adequately stable and that correspond closely to population factors should be at least 10 times the number of items. [19]. As the THWRQLS has 32 items, a sample size of 320 was used. Three statistical criteria were also used. The first was discrimination—the power of an item to separate respondents with good and poor quality of work-life, which was analyzed using a t-test or Mann–Whitney U test. The second was the item-total correlation, evaluated using the correlation among the scores for each item. The average of the correlation of the remaining tests that were still candidates for inclusion in the measurement was more than 0.3 [20]. The third criterion was based on exploratory analysis, which aimed to reduce the number of items in the questionnaire while retaining scale reliability. A principal component analysis (PCA) was carried out on the THWRQLS data set, and a low loading variable factor reduction process was used to reduce the number of variables in the original scale. Items with a loading of less than 0.4 were removed from the item set [21].

Judgmental criteria consisted of expert evaluation of content validity and agreement in terms of item selection. Five experts who had experience in the psychometric properties of the questionnaire were included in the study. The content validity was measured using the item content validity index (I-CVI), with a score of 0.8 or higher, indicating excellent content validity [22]. Agreement regarding item selection was determined using a four-point Likert scale. Items were included in the brief THWRQLS if there was strong agreement, as indicated by an average score greater than 3. A conference was then arranged to allow the experts to reach a consensus on the item selection.

### 2.2. Validity and Reliability of the Brief THWRQLS

#### 2.2.1. Participants and Data Collection

Health personnel at the Khon Kaen University (KKU) Faculty of Medicine (Khon Kaen, Thailand) completed a self-administered online survey based on the brief THWRQLS from April 1 to June 30, 2019. To be included in this study, the participants had to have been working at the KKU Faculty of Medicine for at least one month. Health personnel who were on sabbatical leave were excluded. We calculated our minimum sample size as 10 times the number of items in the scale [19]. Considering that the brief THWRQLS contained 25 items and factoring in a 30% expected loss, we determined that 360 sets of questionnaires would need to be distributed to the target health personnel. After the selection of components to be included in the brief THWRQLS, the online website questionnaire was developed by a programmer. The self-administration online questionnaire contained two parts, the brief THWRQLS, and personal and occupational information. Participants with incomplete answers were not able to submit answers.

#### 2.2.2. Data Analysis

Stata 15 (StataCorp, USA) was used for data analysis. A confirmatory factor analysis was performed to analyze the construct validity of the scale. Standardized factor loading and the data-model fit evaluation based on the Chi-square goodness-of-fit index, Comparative Fit Index (CFI), Root Mean Square Error of Approximation (RMSEA), and Standardized RMR (SRMR) were used. Cronbach’s alpha values were calculated to test the internal consistency. A Cronbach’s alpha of greater than 0.7 was considered as consistent [23]. However, a Cronbach’s alpha of 0.6 was acceptable in this study because there were fewer items in some dimensions than in others.

## 3. Results

### 3.1. Development of the Brief THWRQLS

Statistical and judgmental criteria were used to eliminate items from the scale. Statistical criteria were based on discrimination analysis, item-total correlation, and exploratory factor analysis. All items in the THWRQLS were statistically significant (*p* <0.01) according to discrimination analysis, demonstrating that they had discrimination power. In terms of item-total correlation, only SAW3 had a Pearson correlation of less than 0.3, indicating that it should be eliminated from the original scale. For the exploratory factor analysis, a seven-factor model was generated from the principal component analysis. Twelve items (CAW3, HWI4, GWB1, GWB3, GWB5, GWB6, JCS4, JCS5, JCS6, WCS3, WCS4, and SAW3) had factor loadings of less than 0.4, and it was therefore decided that they should be eliminated from the item set (Table 1). Judgmental criteria consisted of content validity and agreement on item selection. In terms of content validity, four items (EET1, HWI2, HWI3, and GWB4) had I-CVI of less than 0.8, indicating that they should be eliminated. Regarding agreement on item selection, five items (EET1, CAW3, GWB1, GWB3, and SAW4) had an average score of less than 3, making them candidates for elimination from the original scale (Table 1).

Item selection was based on expert consensus. The results of the analyses above were presented to the experts to help them with the decision-making process. The result was the elimination of EET1, CAW2, HWI3, GWB1, GWB3, WCS3, and SAW2 from the THWRQLS (Table 1), leaving seven dimensions with 24 items and one overall item in the final scale. The content of each item in brief THWRQLS were shown in Appendix A.

### 3.2. Validation and Reliability of the Brief THWRQLS

#### 3.2.1. Characteristics of the Participants

Out of 350 invited employees, 250 completed the questionnaire (response rate 71.4%). Forty-six percent of the 250 participants were between 20 and 30 years old. Two hundred and eleven (84.4%) were female, 53.2% were single, and 63.2% had at least one underlying disease. Most were nurses and practitioners (67.6% and 92.8%, respectively). Most of the participants (64.8%) worked 50 to 100 hour per week, and 45.2% had work experience of less than or equal to 5 years (Table 2).

#### 3.2.2. Construct Validity of the Brief THWRQLS

Confirmatory factor analysis for the seven-factor, 24-item model was conducted in all participants and revealed four items (bGWB1, bWCS2, bSAW1, bSAW3) to be problematic, with standardized factor loadings below 0.50. Standardized factor loadings across the 24 items ranged from 0.37 to 0.88 (Table 3). We determined that the model shown in Figure 1 provided an acceptable fit based on its goodness-of-fit statistics (Chi-square = 268.77, *p*-value < 0.01, CFI = 0.97, RMSEA = 0.04 and SRMR = 0.05).

#### 3.2.3. Internal Consistency of the Brief THWRQLS

The internal consistency of the overall scale and subscale based on this model were satisfactory. The calculated Cronbach’s alpha value for the seven subscales ranged from 0.53 to 0.8, and that of the overall scale was 0.94 (Table 4).

## 4. Discussion

The purpose of this study was to develop and evaluate the psychometric characteristics of the brief THWRQLS. The selection of THWRQLS components was conducted based on expert consensus using statistical and judgmental criteria and thus relied on both statistical analysis and expert decision-making. The advantage of using statistical criteria is that they can be used to make quantitative comparisons [24]. However, the results of statistical analysis conducted using data from physicians may not be generalizable to other occupations. In addition, statistical analysis does not take the content of the question into account. Expert opinions based on knowledge, experience, and evidence, on the other hand, are able to reliably explain work-life discrepancies [25]. The appointed experts’ study decided to eliminate seven items which did not meet the criteria, were overly similar to other items, were unclear, or were not related to the dimensions in which they were included.

Test validity is important in the development of any tool because it indicates the usefulness and effectiveness of each component of the scale [26]. The scale’s construct validity was tested using confirmatory factor analysis. We found that there was one item in the “general well-being” dimension, one item in the “working conditions” dimension, and two items in the “stress at work” dimension that had standardized factor loadings of less than 0.5 which may be due to most items being negatively framed or dealing with sensitive issues, which could result in people giving neutral responses to avoid expressing their true opinions or feelings [27] and therefore affecting the validity or reliability of the questionnaire. The other reason is that this scale is multidimensional. The exploratory factor analysis can answer how many dimensions should consist of this scale [28]. However, dividing into seven dimensions is useful to assess each problem to solve the problem on each dimension. Confirmatory factor analysis found this to be a well-fitting model, but there were many pairs of items with high covariance that could be grouped in the same dimension to reduce the overall number of dimensions. The internal consistency of the overall scale and subscale based on this model were satisfactory. The Cronbach’s alpha coefficient of the scale overall was 0.94, and most of the dimensions were greater than 0.7. The one exception was “stress at work,” which had a Cronbach’s alpha coefficient of 0.53. A low value of alpha could be due to a low number of questions, with poor correlation between items meaning that then some should be revised or discarded. The easiest method to find them is to compute the correlation of each test item with the total score test, deleting items with low correlations (approaching zero) [23].

## 5. Conclusions

The brief THWRQLS was shown to have appropriate psychometric properties, as evidenced by the construct validity and internal consistency and can, therefore, be used to evaluate the quality of work-life of healthcare personnel. Besides, the brief THWRQLS was divided into seven dimensions, allowing the results of this assessment to be used to solve each problem. Finally, some limitations remain. This study was conducted with health personnel, who may not be a representative sample. It needs to proceed to further research with other occupations. In future, the brief THWRQLS should be used widely to assess the quality of working life, to understanding problems of the questionnaire and the opinions of the users.

## Figures and Tables

**Figure 1 ijerph-17-01503-f001:**
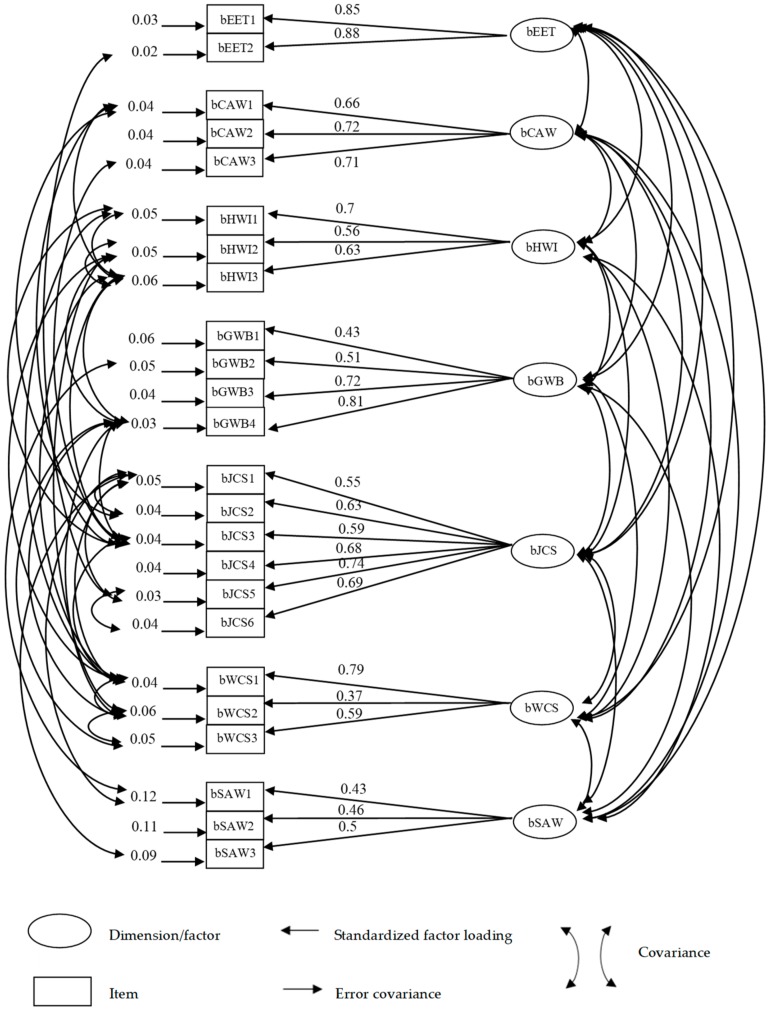
Confirmatory factor analysis of the seven-factor, 24-item brief THWRQLS model (*n* = 250). goodness-of-fit statistics for the structure model: χ^2^ = 268.77, *p*-value < 0.01, CFI = 0.97, RMSEA = 0.04, SRMR = 0.05.

**Table 1 ijerph-17-01503-t001:** Item selection of brief Thai version of the Work-Related Quality of Life Scale (THWRQLS).

Items of THWRQLS	Statistical Criteria			Judgmental Criteria		Items of Brief THWRQLS
	Discrimination (*p*-Value)	Item-Total Correlation	Factor Loading	I-CVIs	Expert Agreement	
EET 1	<0.01	0.52	0.59	0.4	2.8	-
EET 2	<0.01	0.72	0.74	1	4	bETT 1
EET 3	<0.01	0.74	0.66	1	3.8	bETT 2
CAW 1	<0.01	0.62	0.59	1	3.8	bCAW 1
CAW 2	<0.01	0.65	0.55	1	3.8	bCAW 2
CAW 3	<0.01	0.56	0.35	0.8	2.8	-
CAW 4	<0.01	0.65	0.52	0.8	3.6	bCAW 3
HWI 1	<0.01	0.72	0.59	1	3.6	bHWI 1
HWI 2	<0.01	0.69	0.63	0.6	3.2	bHWI 2
HWI 3	<0.01	0.73	0.44	0.6	3	-
HWI 4	<0.01	0.64	0.39	1	4	bHWI 3
GWB 1	<0.01	0.76	0.36	0.8	2.6	-
GWB 2	<0.01	0.62	0.64	1	3.6	bGWB 1
GWB 3	<0.01	0.73	0.32	1	2.8	-
GWB 4	<0.01	0.72	0.44	0.6	3	bGWB 2
GWB 5	<0.01	0.75	0.37	0.8	3.6	bGWB 3
GWB 6	<0.01	0.79	0.36	0.8	4	bGWB 4
JCS 1	<0.01	0.41	0.59	1	3.8	bJCS 1
JCS 2	<0.01	0.58	0.73	1	3.2	bJCS 2
JCS 3	<0.01	0.54	0.51	0.8	3.2	bJCS 3
JCS 4	<0.01	0.66	0.27	1	3	bJCS 4
JCS 5	<0.01	0.72	0.29	1	3.6	bJCS 5
JCS 6	<0.01	0.62	0.31	0.8	3.2	bJCS 6
WCS 1	<0.01	0.63	0.53	1	3.4	bWCS 1
WCS 2	<0.01	0.65	0.63	1	4	bWCS 2
WCS 3	<0.01	0.78	0.28	1	3.4	-
WCS 4	<0.01	0.66	0.37	0.8	4	bWCS 3
SAW 1	<0.01	0.47	0.59	1	4	bSAW 1
SAW 2	<0.01	0.50	0.78	1	4	-
SAW 3	<0.01	0.24	0.37	0.8	3.2	bSAW 2
SAW 4	<0.01	0.39	0.55	0.8	2.8	bSAW 3
OVL	<0.01	0.82	-	0.8	3.6	bOVL

Note. EET = employee engagement, CAW = control at work, HWI = home-work interface, GWB = general well-being, JCS = job and career satisfaction, WCS = working conditions, SAW = stress at work and OVL = overall.

**Table 2 ijerph-17-01503-t002:** Characteristics of the participants.

Characteristics		Frequency (n = 250)	%
Sex	Male	39	15.6
	Female	211	84.4
Age (y)	20–30	115	46.0
	31–40	59	23.6
	41–50	40	16.0
	51–60	36	14.4
Marital status	Single	133	53.2
	Married	99	39.6
	Divorced/Separated/Widowed	18	7.2
Underlying disease	No	92	36.8
	Yes	158	63.2
Profession	Doctor/Dentist	48	19.2
	Nurse	169	67.6
	Pharmacist	14	5.6
	Others	19	7.6
Work role	Leaders	18	7.2
	Practitioners	232	92.8
Years of work	Up to 5 years	113	45.2
	>5 to 15 years	80	32.0
	>15 years	57	22.8
Working hours	Up to 50 h per week	47	18.8
	>50 to 100 h per week	162	64.8
	>100 h per week	41	16.4

**Table 3 ijerph-17-01503-t003:** Standardized factor loading based on confirmatory factor analysis of the brief THWRQLS.

Items	Confirmatory Factor Analysis		
	Standardized Factor Loading	95% CI	*P*-Value
bEET 1	0.85	0.9–5.46	0.00
bEET 2	0.88	0.92–4.89	0.00
bCAW 1	0.66	0.74–5.6	0.00
bCAW 2	0.72	0.8–5.1	0.00
bCAW 3	0.71	0.8–5.0	0.00
bHWI 1	0.70	0.8–4.84	0.00
bHWI 2	0.56	0.67–3.67	0.00
bHWI 3	0.63	0.74–4.26	0.00
bGWB 1	0.43	0.54–3.28	0.00
bGWB 2	0.51	0.6–3.84	0.00
bGWB 3	0.72	0.79–5.58	0.00
bGWB 4	0.81	0.86–5.0	0.00
bJCS 1	0.55	0.64–5.82	0.00
bJCS 2	0.63	0.72–5.96	0.00
bJCS 3	0.59	0.68–4.83	0.00
bJCS 4	0.68	0.75–4.87	0.00
bJCS 5	0.74	0.8–5.2	0.00
bJCS 6	0.69	0.77–5.45	0.00
bWCS 1	0.79	0.86–4.74	0.00
bWCS 2	0.37	0.5–4.16	0.00
bWCS 3	0.59	0.69–4.3	0.00
bSAW 1	0.43	0.66–3.68	0.00
bSAW 2	0.46	0.67–3.41	0.00
bSAW 3	0.50	0.68–3.47	0.00

Note. bEET = employee engagement (brief), bCAW = control at work (brief), bHWI = home-work interface (brief), bGWB = general well-being (brief), bJCS = job and career satisfaction (brief), bWCS = working conditions (brief) and bSAW = stress at work (brief).

**Table 4 ijerph-17-01503-t004:** Internal consistency of the brief THWRQLS.

Factor	Cronbach’s Alpha
bEET	0.8
bCAW	0.73
bHWI	0.74
bGWB	0.8
bJCS	0.78
bWCS	0.71
bSAW	0.53
Overall Scale	0.94

Note. bEET = employee engagement (brief), bCAW = control at work (brief), bHWI = home-work interface (brief), bGWB = general well-being (brief), bJCS = job and career satisfaction (brief), bWCS = working conditions (brief) and bSAW = stress at work (brief).

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
