# Peer review of "Validation of the Brief Thai Version of the Work-Related Quality of Life Scale (Brief THWRQLS)"

_ijerph, 2020, doi:10.3390/ijerph17051503_

Round 1
Reviewer 1 Report
Manuscript ID: ijerph-723990
Manuscript Title: Validation of the Brief Thai Version of the Work-Related Quality of Life Scale (Brief THWRQLS)
Comments
This manuscript proposes to develop and evaluate the validity of a brief Thai version of the work-related quality of life scale. The Introduction section is well written, with a clear background and research question. The Methods section is also clearly reported, with procedures consistent with the study aims. Results are sound and are presented in coherence with the applied methods. I have only minor comments for the authors to consider or rebut.
Abstract
Possible typo (‘…A[a]lmost [all] dimensions…’)
Methods
Line 106. This information is repeated from lines 80-81.Lines 107-108: Just a quick note on the estimated sample size. Distributing 350 questionnaires with an expected loss of 40% would recall only (350*[1-0.4]=) 210 questionnaires. Your 40% loss is somewhat reflected at the 28.6% loss, returning 250 questionnaires. To obtain the desired 320 questionnaires with the expected 40% loss would require distributing the questionnaire to (320/[1-0.4]=) 533 participants. Recalculating using the reported loss, a total of 448 participants should be contacted. Probably not a major issue, but it is worth correcting the calculation for the report.
Discussion
Line 178: This information is repeated from lines 80-81 and line 106. Nonetheless, there is no discussion regarding the enrolled sample size being smaller than the required (320).Author Response
Response to Reviewer 1 Comments
Point 1: Possible typo (‘…A[a]lmost [all] dimensions…’)
Response 1: corrected to almost all dimensions (lines 27).
Point 2: Line 106. This information is repeated from lines 80-81.
Response 2:
- Lines 80-81 describe to the minimum sample size necessary to exploratory analysis (The third criterion of item selection in the brief THWRQLS).
- Line 107 (Original line 106) describe to the minimum sample size necessary to confirmatory analysis (The construct validity of the brief THWRQLS).
Point 3: Lines 107-108: Just a quick note on the estimated sample size. Distributing 350 questionnaires with an expected loss of 40% would recall only (350*[1-0.4]) =210 questionnaires. Your 40% loss is somewhat reflected at the 28.6% loss, returning 250 questionnaires. To obtain the desired 320 questionnaires with the expected 40% loss would require distributing the questionnaire to (320/ [1-0.4]) =533 participants. Recalculating using the reported loss, a total of 448 participants should be contacted. Probably not a major issue, but it is worth correcting the calculation for the report.
Response 3: The minimum sample size necessary to confirmatory analysis (The construct validity of the brief THWRQLS) was 250 participants (10 times the 25 items in brief THWRQLS) and corrected to 30% expected loss, we determined that 360 questionnaires would need to be distributed.
Point 4: Line 178: This information is repeated from lines 80-81 and line 106. Nonetheless, there is no discussion regarding the enrolled sample size being smaller than the required (320).
Response 4: Line 187-188 (Original line 178) deleted the repeated information and the sample size necessary to confirmatory analysis was 250 participants (10 times the number of items in brief THWRQLS). 320 participants were minimum sample size necessary to exploratory analysis (The third criterion of item selection in the brief THWRQLS).

Reviewer 2 Report
The research in this article has certain social application value. The adopted technical methods appear scientifically sound to me.
Line138:The description of the questionnaire survey is relatively simple. Can you supplement it? such as: the source of the questionnaire, the survey method, how to send the questionnaires of the participants? How to ensure the correctness of the questionnaire?
Line163: How do you think the overlapping items in the 32-item THWRQLS have an impact on the psychological underperformance? Compared to the original table, how does the abbreviated scale you are trying to develop reflect the advantages?
Line203: The conclusions of this article are too simple, and it is recommended to mine the complete conclusions. Several aspects of the short THWRQLS are recommended to be described in the previous section.
What are the applicability and limitations of the short THWRQLS in different occupations? The seven aspects of the short THWRQLS seem to be universal, so why not apply it directly to other professions?
Author Response
Response to Reviewer 2 Comments
Point 1: Line138:The description of the questionnaire survey is relatively simple. Can you supplement it? such as: the source of the questionnaire, the survey method, how to send the questionnaires of the participants? How to ensure the correctness of the questionnaire?
Response 1: Additional sentences were (line 112-114)
- After the selection of components to be included in the brief THWRQLS, the online website questionnaire was developed by a programmer.
- The questionnaires were distributed using QR codes by the researchers.
- Participants with incomplete answers would not be able to submit answers.
Point 2: Line163: How do you think the overlapping items in the 32-item THWRQLS have an impact on the psychological underperformance? Compared to the original table, how does the abbreviated scale you are trying to develop reflect the advantages?
Response 2: The reason for developing a brief THWRQLS was mentioned in the introduction part. Here is a further explanation. “The appointed experts decided to eliminate seven items which did not meet the criteria, were overly similar to other items, were unclear or were not related to the dimensions in which they were included.” (line 184-186)
Point 3: Line203: The conclusions of this article are too simple, and it is recommended to mine the complete conclusions. Several aspects of the short THWRQLS are recommended to be described in the previous section.
Response 3: This part has moved to the conclusion
The brief THWRQLS had appropriate psychometric properties, as evidenced by the construct validity and internal consistency and can, therefore, be used to evaluate the quality of work-life of healthcare personnel. Besides, the brief THWRQLS was divided into seven dimensions, allowing the results of this assessment to be used to solve each problem. Finally, some limitations remain. This study was conducted with health personnel, who may not be a representative sample. It will need further study in a different occupation. In the future, the brief THWRQLS should be widely used to assess the quality of working life. To assess the problem in the questionnaire and the opinions of the user.
Point 4: What are the applicability and limitations of the short THWRQLS in different occupations? The seven aspects of the short THWRQLS seem to be universal, so why not apply it directly to other professions?
Response 4: I agree with your suggestion, but each occupation has different working condition, organization factor and the psychological factor which affects the quality of work life. and further recommendation has already stated in “This study was conducted with health personnel, who may not be a representative sample. It needs to further research on the other occupation.” (line 220-221)

Reviewer 3 Report
Authors have performed a very short manuscript on a brief Thai version of the work-related quality of life scale that would reduce the amount of content without sacrificing measurement precision. Validity and reliability were also assessed. I believe the manuscript is well written, straightforward, and contains important data hence I advise publication after addressing minor concerns.
Authors state that previous research suggests that the minimum sample size necessary to obtain factor solutions should be at least 10 times the number of items. I am not a statistician but do authors feel this still applies to todays conditions?
Again I am not a statistician but regarding the choice of Persons correlation and not Spearman, have authors tested for normality of distribution?
Figure 1 needs improving as several aspects are overlaying (arrows and numbers, letters with letters). As it stands it is confusing
Author Response
Response to Reviewer 3 Comments
Point 1: Authors state that previous research suggests that the minimum sample size necessary to obtain factor solutions should be at least 10 times the number of items. I am not a statistician but do authors feel this still applies to today’s conditions?
Response 1: In the factor analysis literature (MacCallum RC, Widaman KF, Zhang S, Hong S), the minimum sample size necessary to obtain factor solutions that adequately stable and that correspond closely to population factors should be at least 10 times the number of items. (line 80-82)
Point 2: Again I am not a statistician but regarding the choice of Persons correlation and not Spearman, have authors tested for normality of distribution?
Response 2: Normality testing was done found that the data were both normal and abnormal distribution therefore, I used “correlation” instead of “Pearsons correlation”. I presume this statistic is correct.
Point 3: Figure 1 needs improving as several aspects are overlaying (arrows and numbers, letters with letters). As it stands it is confusing
Response 3: The symbols are explained as suggested. (Figure 1)

This manuscript is a resubmission of an earlier submission. The following is a list of the peer review reports and author responses from that submission.